# Genome-Wide Analysis of Cyclic Nucleotide-Gated Channel Genes Related to Pollen Development in Rice

**DOI:** 10.3390/plants11223145

**Published:** 2022-11-17

**Authors:** Su-Kyoung Lee, Soo-Min Lee, Myung-Hee Kim, Soon-Ki Park, Ki-Hong Jung

**Affiliations:** 1Graduate School of Green-Bio Science and Crop Biotech Institute, Kyung Hee University, Yongin 17104, Republic of Korea; 2School of Applied Biosciences, Kyungpook National University, Daegu 41566, Republic of Korea

**Keywords:** *Oryza sativa*, CNGC, pollen germination, pollen tube growth, yeast two-hybrid

## Abstract

In the angiosperm, pollen germinates and rapidly expands the pollen tube toward the ovule. This process is important for plant double fertilization and seed setting. It is well known that the tip-focused calcium gradient is essential for pollen germination and pollen tube growth. However, little is known about the Ca^2+^ channels that play a role in rice pollen germination and tube growth. Here, we divided the 16 cyclic nucleotide-gated channel (CNGC) genes from rice into five subgroups and found two subgroups (clades II and III) have pollen-preferential genes. Then, we performed a meta-expression analysis of all *OsCNGC* genes in anatomical samples and identified three pollen-preferred *OsCNGCs* (*OsCNGC4, OsCNGC5*, and *OsCNGC8*). The subcellular localization of these OsCNGC proteins is matched with their roles as ion channels on the plasma membrane. Unlike other *OsCNGCs*, these genes have a unique cis-acting element in the promoter. OsCNGC4 can act by forming a homomeric complex or a heteromeric complex with OsCNGC5 or OsCNGC8. In addition, it was suggested that they can form a multi-complex with Mildew Resistance Locus O (MLO) protein or other types of ion transporters, and that their expression can be modulated by Ruptured Pollen tube (RUPO) encoding receptor-like kinase. These results shed light on understanding the regulatory mechanisms of pollen germination and pollen tube growth through calcium channels in rice.

## 1. Introduction

For successful reproduction and seed formation in the angiosperm, sperm cells should be transported to the ovules. When pollen lands on the stigma, the compatible pollen germinates, and the pollen tube elongates along the style. Pollen tube growth is a polarized cell expansion process involving a directional mechanism in response to guidance signals, and Ca^2+^ signals have been demonstrated to be involved in polarized cell growth such as pollen, root hair, neurons, and fungi [1,2]. Especially, Ca^2+^ has an essential role in pollen germination and pollen tube growth, and its distribution in the pollen tube affects pollen tube guidance and tip elongation [3,4]. Pollen tube growth depends on tip-focused Ca^2+^ gradients and the spatial modification of the Ca^2+^ gradients causes the reorientation of pollen tube growth [5]. The Ca^2+^ gradient in the pollen tube is produced by external Ca^2+^ influx, which is mainly mediated by Ca^2+^ channels located at the apical tip of the pollen tube [6]. Thus, Ca^2+^ channels are key components for regulating Ca^2+^ gradients and are essential for pollen tube growth and the directional regulation of tubes.

Recent studies have reported that cyclic nucleotide-gated ion channels (CNGCs) are the potential Ca^2+^ channels causing the change in cytosolic Ca^2+^ concentration during signal transduction [7]. CNGCs are tetrameric cation channels spanning the plasma membrane [8] and their isoforms were identified in plants as Ca^2+^-permeable channels involved in pathogen defense signaling cascades [9], pollen germination and tip growth [10,11], and plant responses to other biotic and abiotic stresses [12]. Animal CNGCs have been well elucidated regarding their biological functions, regulation, and conserved structures. In plants, studies on the molecular function and regulation of plant CNGCs as well as the structure and function of the CNGC family have recently been conducted [13]. It is known that plant CNGC family proteins generally have six transmembrane domains (S1–S6) with a pore domain (P loop) between S5 and S6 that permits ion transport, the C-terminal cyclic nucleotide-binding domain (CNBD), and the CaM-binding domain (CaMBD) [14,15,16]. The major feature of plant *CNGC* gene families is CNBD, which has a phosphate binding cassette (PBC) binding to the sugar and phosphate moieties of the cyclic nucleotide (cNMP, cyclic nucleotide-monophosphate) ligand [17], and a “hinge” region (adjacent to the PBC) essential for the capping of cAMP by the C-helix of the CNBD [18] and for influencing ligand selectivity and binding affinity [19]. 

In plants, genome-wide analyses of *CNGC* gene families have been reported in Arabidopsis [20], rice [7], wheat [21], maize [22], tobacco [23], tomato [24], pear [25], and *Brassica oleracea* [26]. In particular, the cation channels in legumes have been used to elucidate the contribution of the CNGC family to the signal transduction pathways [27]. In Arabidopsis, CNGCs that preferentially express in pollen tissues play a crucial role in the tip growth of pollen tubes and pollen tube guidance [10,11,28,29,30]. Null mutations of *AtCNGC18* resulted in abnormal cytosolic Ca^2+^ oscillation in pollen tube tips and defective pollen tubes that were unable to undergo directional growth and reach the ovule for fertilization [10,11]. *AtCNGC16* is also responsible for pollen tube growth and fertility in heat stress and drought responses [28]. In rice, 16 *CNGC* members have been identified, and the function of *OsCNGC13*, as a maternal sporophytic factor for Ca^2+^ gradient regulation in the stylar, has been reported [31]. However, there are no reports studying *CNGC*, which functions in pollen development and can act as a paternal factor.

Genome-wide analysis of the CNGC family in *Oryza sativa* (rice) and phylogenetic analyses with Arabidopsis CNGC proteins was performed using 16 *CNGC* full length genes in rice. Then, three pollen-preferred *CNGC* genes in rice was identified and the expression pattern of these *OsCNGC* genes was analyzed through quantitative polymerase chain reaction (qPCR) analysis to verify their function related to pollen developments. In addition, subcellular localization analysis of these OsCNGC proteins was performed via a protoplast isolation system, and putative pollen-preferred cis-acting regulatory elements (CREs) were identified through a comparative CRE analysis using *OsCNGC* gene promoters. Finally, the CNGC-mediated regulatory network for pollen germination and tube elongation processes was generated. 

## 2. Materials and Methods

### 2.1. Motifs Analysis

The CNGC protein sequences of rice and Arabidopsis were subjected to the Multiple Expectation MEME version 5.4.1 analysis online (https://meme-suite.org/meme/index.html (Accessed on 1 May 2022)). To investigate the protein motifs in detail, the five groups of CNGCs were analyzed independently. A total of 30 sequences with a minimal width of 6 aa and a maximal width of 250 aa were found.

### 2.2. Meta-Expression Analysis

To examine expression patterns, the Affy package encoded by the R language was used to normalize the signal intensity and then the data were transformed into log2 values of expression fold-change. Microarray data, including Affymetric and Agilent array data, were downloaded from the NCBI GEO (https://www.ncbi.nlm.nih.gov/geo/ (Accessed on 15 February 2022)) and Genevestigator (https://genevestigator.com/gv/ (Accessed on 15 February 2022)). Then, the normalized data were uploaded to the Multi Experiment Viewer 6 and the data were visualized using heatmaps. The levels of gene expression in several organs were compared using Genvestigator and the functional similarity among rice and Arabidopsis members was estimated.

### 2.3. Multiple Sequence Alignment and Phylogenetic Tree Construction

To perform the phylogenetic analysis of the CNGC proteins in rice and Arabidopsis, protein sequences with a locus ID were collected from the Rice Genome Annotation Project (http://rice.uga.edu/ (Accessed on 18 January 2022)), NCBI, and Phytozome platform (https://phytozome-next.jgi.doe.gov/ (Accessed on 18 January 2022)). Multiple alignments of the amino acid sequences were performed using ClustalW. The phylogenetic analysis was performed using MEGA 7.0.26 with maximum likelihood and neighbor-joining methods (bootstrap repeat value was 1000), as previously described [32].

### 2.4. Cis-Acting Element Analysis

To identify the pollen-preferred cis-elements in the promoter sequences of three *OsCNGC* genes, the 2 kb upstream sequence was searched for in the Plant cis-acting regulatory DNA elements (PLACE) database [33]. To examine the candidate regions for CREs, the motif finding program MEME version 5.4.1 was used [34]. Cis-acting element analysis was performed by comparing the 2 kb upstream sequences from the start codon of three pollen-preferred *OsCNGC* genes, as previously described. Ten sequences with a minimal width of 5 bp and a maximal width of 10 bp were searched. The motifs discovered were further analyzed using the TomTom tool and those found only in the promoter sequence of pollen-preferential *CNGC* genes were determined. GO analysis of transcription factors that bind to known CREs was analyzed using the Rice Oligo Array Database (http://ricephylogenomics-khu.org/ROAD_old/analysis/go_enrichment.shtml (Accessed on 1 May 2022)).

### 2.5. RNA Extraction, cDNA Construction, and qRT-PCR Analysis

Tissue samples, including pollen of rice plants (*Oryza sativa* cv. Dongjin) grown in paddy fields, were frozen in liquid nitrogen and homogenized with a TissueLyser II (Qiagen, Hilden, Germany). Total RNA was extracted using RNAiso Plus, as described previously [35]. cDNAs were synthesized using a SuPrimeScript RT premix from GeNet Bio (Chungcheongnam-do, South Korea) [36]. qRT-PCR was performed with a Qiagen Rotor-Gene Q qRT-PCR cycler (Qiagen) using the following thermal cycling procedure: 95 °C for 30 s, 40 cycles of 95 °C denaturation for 10 s and 60 °C annealing for 30 s, and 72 °C extension for 1 min. To evaluate tissue-preferred expression patterns by qRT-PCR, Rice ubiquitin 5 (*OsUbi5*, *LOC_Os01g22490*) [37] was used as an endogenous control to normalize the variance in the amount of sample cDNAs. Gene-specific qRT-PCR primers were designed for a specific region of each gene, as described in Appendix A, and the accuracy and efficiency of each primer set was verified through PCR amplification of the gDNA to optimize the PCR conditions and melting curve. The fold-change in expression was calculated using the comparative delta-CT [38,39]. Three biological replicates were analyzed for each sample, with at least three technical replicates of those measured.

### 2.6. Subcellular Localization Analysis

The coding sequences (CDSs) of three *CNGC* genes were amplified from mature anther cDNA and cloned into pGA3574 vectors fused with C-terminal GFP. All the cloning primers used in the experiments are listed in Appendix A. For the transfection of these constructs, rice protoplasts from leaf and stem tissue were isolated, as described previously [40]. The fluorescent tags used to determine the intracellular distribution of proteins were observed using a confocal laser scanning fluorescence microscopy (K1-Fluo; Nanoscope System, Daejeon, Republic of Korea) with 488/510 nm excitation/emission filter sets. To identify the subcellular localization in the cell membrane, the plasma and vacuolar membrane marker FM4-64 (Thermo Fisher Scientific, Waltham, MA, USA) was used; rice protoplasts were immersed in 0.1% FM4-64 solution for 10 min in the dark conditions and observed using the red fluorescence protein channel with 543/560 nm excitation/emission filter sets.

### 2.7. Yeast Two-Hybrid Analysis

The C-terminal CDS of *OsCNGC4* (1654–end), *OsCNGC5* (1741–end), and *OsCNGC8* (1288–end) were cloned at the *Eco*RI/*Bam*HI site of the pGADT7 vector and the pGBKT7 vector [41]. To test the self-transcriptional activity of the OsCNGC bait, the plasmid was introduced with an empty prey vector into the yeast strain AH109. Yeast transformants of the bait and prey plasmids were selected on an SD minimal medium lacking leucine and tryptophan (SD-LW) and were replica-plated onto various SD selection media, including medium lacking leucine, tryptophan, and histidine (SD-LWH, +5 mM 3-AT) and medium lacking leucine, tryptophan, and adenine (SD-LWA).

### 2.8. Network Analysis

The protein interaction network was predicted by the STRING database version 11.5 (https://string-db.org/ (Accessed on 10 June 2022)), which obtains interactions based on genomic, experimental, co-expression or previous knowledge information context at the function or physical level. String network analysis was performed using the protein sequences of 16 *OsCNGC* genes.

## 3. Results

### 3.1. Phylogenetic Tree and Motif Composition Analyses of OsCNGC Proteins

To reveal the evolutionary relationships of CNGC proteins in plant species, the amino acid sequences of 16 CNGC proteins in rice and 20 CNGC proteins in Arabidopsis were identified and aligned (Figure 1A, Appendix A). Subsequently, the proteins were clustered into five groups, as described by Mäser et al. [20]. The rice and Arabidopsis *CNGC* genes were not classified according to species, and similar numbers of genes were grouped together in each cluster, implying that rice and Arabidopsis *CNGC* gene family members appear to have coevolved into monocots and dicots. Clade I comprised *AtCNGC1, 3, 10, 11, 12,13, OsCNGC1, 2*, and *3*; clade II comprised *AtCNGC5, 6, 7, 8, 9, OsCNGC4, 5*, and *6*; clade III comprised *AtCNGC14, 15, 16, 17, 18, OsCNGC7, 8, 9, 10*, and *11*; clade IVa comprised *AtCNGC19, 20, OsCNGC12*, and *13*; clade IVb comprised *AtCNGC2, 4, OsCNGC14, 15*, and *16*. As *AtCNGC7, 8, 16*, and *18* exhibited pollen-preferred expression in Genevestigator transcriptome data (Appendix A), *OsCNGC4, 5*, and *6* are orthologs of *AtCNGC7* and *8*, whereas *OsCNGC8* is an ortholog of *AtCNGC16* and *18*. In addition, according to The Arabidopsis Information Resource (TAIR, https://www.arabidopsis.org/ (Accessed on 5 August 2022)), the orthologs of *AtCNGC7* with key roles in pollen development are *OsCNGC4, 5,* and *6*, which means that *OsCNGC4, 5*, and *6* may perform similar functions to *AtCNGC7* and *AtCNGC8*.

To examine the structural conservancy or divergence among the *CNGC* genes in rice and Arabidopsis, the conserved motifs in the CNGC proteins were investigated using tools from the Maximization for Motif Elicitation (MEME) suite. The five groups of CNGC were analyzed separately (Figure 2A, Appendix A). With the exception of the lack of motifs 25, 23, and 28 at the N-terminal and motifs 24 and 30 at the C-terminal, clade I possessed most of the motifs contained in the other clades, and the differential motif patterns of the protein structures varied relatively according to the clade. In particular, six motifs in CNBD of clade I, motifs 2, 6, 13, 16, 19, and 22, were present in almost all clades. 

It can be suggested that members of clade I have emerged as a genetic basis for evolution to form CNGC subfamilies in different directions. Similar to clade I, CNGCs in clades II and III underwent partial differentiation at the same location. However, the N-terminals and C-terminals from clades IVa and IVb have unique motifs, such as motifs 23, 28, 29, and 24, which do not exist in clades I, II, and III. In case of clade Ⅳa, the CNBD domain was finally differentiated as to have motifs 19, 16, and13 and in case of clade Ⅳb, to have motifs 6, 16, and 13. This may suggest that the members of clade Ⅳ have evolved in different directions to have different functions with the other groups.

Among the motifs preserved at the same location in all clades, 11 motifs with known functions are defined as the characteristic features of the CNGC family (Figure 2B). The seven motifs preserved in the N-terminal of the CNGC family are involved in the formation of the regions of the transmembrane and pore, which have been implicated in ion selective transport. In addition, CNGCs are characterized by the presence of CNBD, namely LI-X(2)-[GS]-X-[FV]-X-G-[1]-ELL-X-W-X(12,22)-SA-X(2)-T-X(7)-[EQ]-AF-XL, at the C-terminus. The CNGC families of both rice and Arabidopsis contain CNBD in the C-terminal region (Figure 1B). The most conserved residues in the PBC within the C-terminus, which is important for cyclic AMP binding [42], was also located in *OsCNGC* genes (Appendix A). The CNBD was known to be composed of four α-helices and eight β-sheets [43]. The secondary structure of the *OsCNGC* gene was confirmed using amino acid sequences and the presence of CNBD, consisting of four α-helices and eight β -sheets at the C-terminal, was also confirmed (Appendix A). In addition, we found that motif2, reported as the IQ domain, is preserved at the C-terminus of all clades of CNGC, and its preserved core signature corresponds to [A-x(3)-I-Q-x(2)-F-R-x(4)-K] in all CNGC members of rice. The IQ domain was known as the CaM-binding site within CaMBD [14], which is another representative domain among plant CNGC families. Apo-CaM (CaM without Ca^2+^ ligands) constitutively binds to the IQ domain in a Ca^2+^-independent manner to act as a Ca^2+^ sensor [14] and many IQ motifs are identified in protein kinase C (PKC) phosphorylation sites [44,45].

### 3.2. Expression Pattern Analysis of OsCNGC Genes

To identify genes that may play a role in the pollen developmental stage, the Rice Male Gamete Expression Database was used, which provides meta-expression analysis focused on rice male gamete development [46]. Then, by aligning the OsCNGC protein sequences, we constructed a phylogenic tree of the rice CNGC family to check the function of *OsCNGCs* related to pollen development (Figure 3A). Meta-expression data from the publicly available Affymetrix rice microarray data sets (National Center for Biotechnology Information [NCBI] GEO; http://www.ncbi.nlm.nih.gov/geo/ Accessed on 15 February 2022)) were also used to check the expression of these genes in various tissues/organs (Appendix A) [47]. We found that the expression patterns of three genes (*OsCNGC4, OsCNGC5, OsCNGC8*) were closely associated with late pollen development at the bicellular, tricellular, mature, and germinated pollen stages. Two genes, *OsCNGC4* and *OsCNGC5* in clade II, which were located in the same sister node of the phylogenetic tree, were preferentially expressed in pollen tissues, whereas *OsCNGC6*, which belongs to the same sister node, was ubiquitously expressed in many tissues. *OsCNGC8* in clade III was only highly expressed in the late pollen stage. Thus, we could expect that these three genes (*OsCNGC4, OsCNGC5*, and *OsCNGC8*) might be associated with pollen function. 

To verify the meta-expression patterns in the late pollen development stages, real-time (qPCR) analysis (Figure 3B) was performed. We used anther tissues at various developmental stages containing meiosis microspores, tetrad microspores, young microspores, vacuolated pollen, and mature pollen, with other tissues, such as root, shoot, and seed. The results demonstrated that pollen-preferred *OsCNGC* genes had the highest expression at the later stages, but not in other tissues. These three genes were used for further studies.

### 3.3. Cis-Acting Regulatory Element Analysis for Promoter Regions of OsCNGC Family Genes

Promoters contain various CREs that are responsible for the development and physiology of plants through regulating gene expression [48]. To find the CREs in the promoter sequence that are responsible for pollen-preferred expression, the 2000 bp sequences upstream from the transcription start site of *OsCNGC4, 5*, and *8* showing preferential expression in pollen were scanned using the PLACE database. Major CREs for pollen-preferred expression, such as POLLEN1LELAT52 (AGAAA), GTGANTG10 (GTGA), and PB Core (CCAC), were found and are summarized in Appendix A. However, in the case of the remaining *OsCNGC* family genes, there were also a significant number of known pollen-preferential CREs in the promoters of the other *OsCNGC* genes that exhibited little or no expression in pollen. On average, there were 10.7 copies of POLLEN1LELAT52, 12.3 copies of GTGANTG10, and 21.6 copies of the PB Core in the promoters of the other *OsCNGCs*, compared with the three pollen-preferred genes. We calculated the *p*-values to determine how many of the three highly pollen-preferred *OsCNGC* genes had significantly more CREs relative to the rest of the *OsCNGC* genes, but the *p*-values of all known pollen-preferred CREs did not exceed 0.01. Based on these results, we confirmed that known pollen-preferential CREs may not be responsible for regulating the pollen-preferred expression of *OsCNGC4, 5*, and *8*.

Apart from known pollen-preferential motifs, we attempted to find other putative CREs using the motif analyzing tool MEME. The 2000 bp sequences upstream of the three *OsCNGC* genes were used and two probable CREs, TCTTYCTCC and GCGGMGGCG, were found in pollen-preferred *OsCNGCs* (*OsCNGC4*, *5*, and *8*), while the other *OsCNGCs* had none or fewer motifs (Figure 4A). These two CREs were distributed throughout the 2000 bp upstream sites of the transcription start sites of *OsCNGC4, 5*, and *8*. We expected that these two CREs would be involved in late pollen-preferred expression, indicating their functional involvement in the transcriptional regulation process. Additionally, these elements were also found in other genes already notably highly expressed in the late pollen stage during pollen germination and tube elongation, such as *OsANTH* (*LOC_Os02g07900*), *OsGORI* (*LOC_Os03g52870*) and *RUPO* (*LOC_Os06g03610*) [41,49,50]. Our results indicate that these two CREs might act as a regulator of pollen-preferential genes in rice. Next, these CREs were analyzed using the TomTom tool to find transcription factors that can bind to known CREs in the Arabidopsis genome (Figure 4B). We revealed that these elements were presented in transcription factor genomic sequences, such as Far1-Related Sequence (FRS) and Basic Helix-Loop-Helix (bHLH), AP2/ERF, BASIC PENTACYSTEINE (BPC) transcription factor, indicating that the three *OsCNGC* genes could be regulated during late pollen development by transcription factors such as FRS, bHLH, AP2/ERF, and BPC. 

To explore the biological function of all the transcription factors identified in the above analysis, the TAIR bulk Gene ontology (GO) annotation retrieval tool was used to analyze the GO terms for the transcription factors found (Figure 4C). Of these, 19 GO terms were over-represented in potential transcription factors of the late pollen-preferred genes. Significantly enriched terms were found for GO terms in the biological process category, corresponding to the ethylene-activated signaling pathway (over 100–fold GO enrichment value), cellular response to ethylene stimulus (95.h59), phosphor-relay signal transduction system (73.36), response to ethylene (44.35), phloem or xylem histogenesis (17.26), hormone-mediated signaling pathway (16.57), cellular response to hormone stimulus (14.36), intracellular signal transduction (14.11), cellular response to endogenous stimulus (13.52), positive regulation of transcription, DNA-templated (13.25), positive regulation of RNA biosynthetic process and of nucleic acid-templated transcription (13.2), positive regulation of macromolecule biosynthetic process (13.2), and regulation of transcription, DNA-templated (11.78). These values indicate that ethylene response factors and phosphor-relay signaling and hormone-mediated signaling may play an important role in the transcriptional regulation of the pollen-preferred *OsCNGC4, 5*, and *8* genes.

### 3.4. Subcellular Localization of OsCNGC4, 5, and 8 Proteins

The *OsCNGC* genes are generally held to encode cation channel subunits. These may be expressed in the plasma membrane (PM). However, it was proposed that CNGCs are also located in the mitochondrial, nuclear, and vacuolar organelles [51]. In Arabidopsis, localizations of CNGCs have been previously elucidated using chimeric fluorescent reporter proteins. With respect to the pollen-preferred AtCNGCs, AtCNGC18 was revealed to be located at the PM of growing pollen tip and vesicles [10,15], and AtCNGC7 and AtCNGC8 were found to have similar localization patterns in the endo-membrane compartment, especially in tonoplasts [52]. However, AtCNGC7 was also observed at the PM in the bud site of the growing tube in a more recent study [29]. 

The subcellular localization of CNGCs in rice was predicted using Protein Subcellular Localization Prediction Tool (PSORT) analysis by Nawaz et al. [7]. The results demonstrated that 13 of 16 OsCNGCs (OsCNGC1, 2, 4, 5, 6, 8, 9, 10, 12, 13, 14, 15, and 16) are predicted to be localized at the PM, and the other three (OsCNGC3, 7, and 11) may be localized in the cytoplasm, chloroplast thylakoid membrane, and mitochondrial inner membrane, respectively [7].

To monitor whether three candidate genes associated with pollen development (OsCNGC4, OsCNGC5, and OsCNGC8) were localized to the PM, we constructed a green fluorescent protein of (GFP)-fused OsCNGC proteins and used a protoplast-based transient expression system (Figure 5). The C-terminal fusion of GFP resulted in the PM localization of all three candidates. Further, we used FM4-64 staining as a membrane marker to confirm how to correlate with the GFP signal of OsCNGC proteins [53,54]. When protoplasts were stained with FM4-64, the GFP signals from the OsCNGC proteins overlapped with the signal from the FM4-64 staining. These results indicate that the three pollen-preferred OsCNGC proteins are localized in the PM to exert their functionality.

### 3.5. Network Analysis of CNGC-Mediated Genes for Pollen Development

To predict the regulatory pathway mediated by OsCNGCs, we used all OsCNGC proteins and constructed the network of proteins associated with CNGC-mediated biological processes via STRING database (Figure 6A). The results demonstrated that OsCNGC8 might interact with Mildew Resistance Locus O (MLO) proteins (OsMLO4, 12, and 11), an ABCG transporter called the STR1 protein, and the LRR receptor-like kinase Flagellin-Sensitive 2 (FLS2) protein; moreover, OsCNGC4 and 5 also both interact with the FLS2 protein. The network also demonstrated that the CNGC proteins might interact with several transporters, such as TPKA, which is a K+ ion channel and SUT3, which is a sucrose transporter.

It is previously reported that CNGCs interact with each other to form hetero- or homo-tetramer to provide the functionality [55]. It is also known that the C-terminal domains of plant CNGCs play an important role in binding between the CNGC–CNGC interaction [56]. The yeast two-hybrid assay was used to determine whether CNGC members expressed in the late pollen development stage were able to form interactions with each other (Figure 6B). C-terminal of CNGC4 was interacted with itself and with the C-terminal of CNGC4 or CNGC5. The interaction intensity was strongest for CNGC4–CNGC4, but weakest for CNGC4–CNGC5.

Considering the plasma membrane localization of most plant CNGCs, the receptor kinases or receptor-like kinases (RLKs) may be candidates for kinases that regulate CNGCs. In Figure 6A, 14 OsCNGCs interact with FLS2, an LRR receptor-like kinase, which also suggests a functional association between CNGCs and RLKs. A type of RLKs, the Catharanthus roseus receptor-like kinase (CrRLK1L) family, has been identified as the regulator for sexual reproduction in both male and female gametophytes [57,58]. *RUPO* encodes a receptor-like kinase that is a member of the CrRLK1L subfamily in rice [50]. *RUPO* is expressed preferentially in pollen and interacts with potassium transporters, regulating K+ homeostasis for PT growth and integrity [50]. Using quantitative PCR analysis, the expression level of three *OsCNGCs* (*OsCNGC4, OsCNGC5,* and *OsCNGC8*) was investigated in a *rupo* mutant anther and wild-type anther (Figure 6C). Consequently, we found that *rupo* mutations caused significant suppression in expression levels of three *OsCNGC* genes; however, there was no prediction of physical interaction between the RUPO protein and each of the three OsCNGC proteins. 

## 4. Discussion

Plant CNGC proteins form tetrameric cation channels that regulate various physiological processes. Our study performed a genome-wide analysis of *CNGC* genes in rice and identified three candidate genes that were highly expressed in pollen, which might play a role in late pollen development. Then, their feature domains for protein sequence and subcellular localization were analyzed, and a protein–protein interaction network mediated by *CNGC* family genes in rice was proposed.

Previous studies have suggested that *CNGCs* are involved in numerous biological functions, and that these genes can be functionally distinguished in a clade-dependent manner [23]. As demonstrated by our results, *CNGC* members in *Oryza sativa* and *Arabidopsis thaliana* could be divided into five subgroups, comprising clades I, II, III, IVa, and IVb; however, those genes preferentially expressed in pollen tissues were found in clades II and III. When we investigated functional domains of *OsCNGC* family members, we found that the motif structure of *OsCNGC* genes in clades II and III was well-conserved compared with those in Arabidopsis, suggesting that pollen-preferred *CNGC* genes are evolutionarily related and have evolved to be able to function preferentially in pollen tissues.

All CNGC proteins in our study contained S1–S6 domains with a P loop, PBC, and hinge region. These structural features provide important evidence for their co-evolution between rice and Arabidopsis and suggests that the functions in cNMP ligand and ion penetration are the most primitive functions in plant CNGCs; moreover, that functional differentiation has occurred through the expansion of the family. In addition, we found that all OsCNGC members share an Ile–Gln (IQ) motif at the C-terminal CaMBD. The C-terminal CaMBD with this IQ motif is adjacent to the CNBD and may function as the calcium-sensor able to regulate CNGC in a negative or positive manner [28,59]. Through these domains, CNGC might respond to Ca^2+^ signaling involved in various developmental processes and have evolved as an extended Ca^2+^-dependent channel control mechanism.

The pollen-preferred expression pattern of three *OsCNGC* genes can be explained by the presence of CREs that are not found in other *CNGC* promoters. In our promoter analysis, putative pollen-preferential CREs (TCTTYCTCC and GCGGMGGCG) were identified from MEME analysis. In previous studies, *OsCNGC* was reported to regulate gene expression in response to hormone, biotic, and abiotic stress [7]. Similarly, it was found that pollen-preferential CREs are involved in the promoters of several transcription factors related to hormones such as AP2/ERF and BPC [60,61]. Furthermore, as these CREs are also present in the promoters of transcription factors such as bHLH and Far1, it is suggested that these transcription factors will be potential regulators that modulate the pollen-preferred expression of *OsCNGC* genes. However, our findings from the MEME analysis required further confirmation to identify the functional CREs in promoters for the transcription factors.

In the protein network analysis mediated by rice CNGCs, we discovered that diverse functional proteins, such as kinases and transporters, interact with CNGC proteins in rice. Then, an analysis of subunit interaction among pollen-preferred OsCNGC members was also performed, including OsCNGC4, OsCNGC5, and OsCNGC8, by using a yeast two-hybrid analysis. The results demonstrated that OsCNGC4 directly binds to OsCNGC4, OsCNGC5, and OsCNGC8, respectively. However, there were differences in the intensity of interactions. A recent study proposed a novel mechanism for the regulation of heteromeric channel assembly [56]. It was also demonstrated that plant CNGCs formed tetrameric cation channels among CNGC four subunits to perform different unique functions [15,28,29,31,62]. In Arabidopsis, CNGC subunits that have the ability to regulate pollen tube growth, including AtCNGC18, 7, and 8, combine to form a tetrameric complex with various configurations. AtCNGC7 and 8, which were functionally redundant in the pollen tube growth assay, interact with the functionally antagonistic AtCNGC18 subunit, respectively, and form a heterotetramer [56]. Similarly, by examining the functional relationship between the pollen-preferred OsCNGCs, we could predict that OsCNGC4 has a major role in forming the subunit combination of three pollen-preferred OsCNGCs. Additionally, OsCNGC5, belonging to the same sister node in the phylogenetic tree, may be functionally redundant with OsCNGC4, and OsCNGC8 may antagonize OsCNGC4. The activities of CNGC complexes in plant cells could be affected not only by the combinations of subunits, but also by the number of subunits [63]. Unlike plant CNGC proteins, animal CNGC families are less diversified, and the influence of the subunit combination seems to be exiguous. Therefore, it is an interesting discovery that plants may modulate the activity of CNGC subunits by forming various combinations of tetramers, which may realize an understanding of complex Ca^2+^ signaling mechanisms in plants [13,56]. However, further studies, such as a functional assay of these subunit combinations, might provide the crucial evidence to support our hypothesis.

In animals, phosphorylation is one way to regulate cyclic nucleotide-regulated cation channels [64]. Earlier studies have established that protein kinase inhibitors block the activity of calcium channels in plant cells [65,66], and suggest that protein phosphorylation plays an important role in Ca^2+^ signaling. Considering the PM localization of most plant CNGCs, receptor kinases or receptor-like kinases (RLKs) are likely candidates for the kinases that phosphorylate CNGCs. RUPO is the rice member of the plant-specific receptor-like kinase CrRLK1Ls subfamily, and some CrRLK1Ls are known to play an important role in regulating Ca^2+^-dependent pollen tube growth and the interaction with ion transporters by phosphorylation to regulate ion homeostasis [67]. The mutation of *RUPO* was found to affect the expression of three pollen-preferred *OsCNGCs*. The downstream signaling events of individual CrRLK1L pathways were discovered in recent studies and some CrRLK1L members were involved in Ca^2+^ dynamics and were essential for maintaining PT integrity during polarized tip growth [68]. CrRLK1L transmits phosphorylation signals, so we assumed that this signaling pathway mediated by *RUPO* might be essential for *OsCNGC* to regulate Ca^2+^ dynamics during pollen germination and the tube growth of rice. Collectively, we proposed a hypothetical model regarding the function of OsCNGC candidates that exhibit high expression levels during late pollen development. Our model requires a further study of the OsCNGC complex functions affecting pollen germination and tube growth.

## 5. Conclusions

Our study aimed to improve the overall understanding of the pollen germination and pollen tube growth mechanism mediated by Ca^2+^ channels, *CNGCs*. There are three pollen-preferred *OsCNGCs* (*OsCNGC4, OsCNGC5*, and *OsCNGC8*) that can serve as Ca^2+^ channels in the development of rice pollen. We suggest that these CNGCs are located in plasma membranes to serve as Ca^2+^ channels, and CNGC4 can act by forming a homomeric or heteromeric complex with OsCNGC5 or OsCNGC8. This result may mean that the activity of subunits can be controlled through various combinations of CNGCs in rice. In addition, it has been proposed that CNGCs can form complexes with Mildew Resistance Locus O (MLO) proteins, which direct the positioning of CNGCs [30]. CNGCs-mediated complex also can interact with other types of ion channels, and that activity can be regulated by Ruptured Pollen tube (RUPOs) encoding receptor-like kinases. These results can help to understand pollen germination and pollen tube growth mechanisms regulated by mediating the calcium channels in rice. Furthermore, it is meaningful to propose a suitable candidate member for improving crop productivity beyond understanding the molecular mechanisms of pollen germination and pollen tube growth.

## Figures and Tables

**Figure 1 plants-11-03145-f001:**
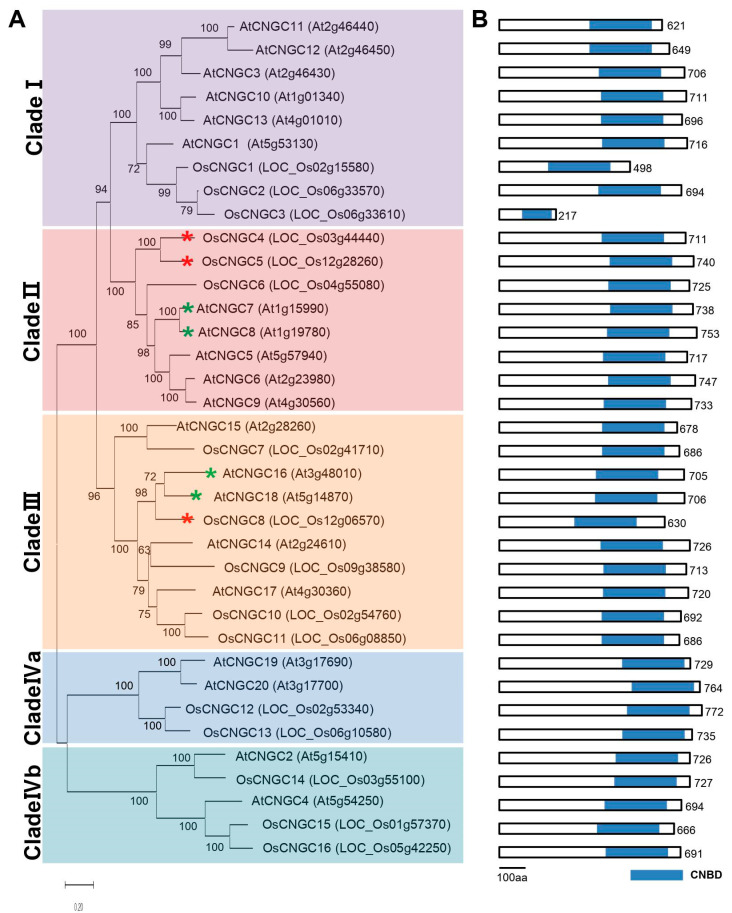
(**A**) The phylogenetic tree was constructed by annotating the protein sequences for each gene by using MEGA7. Red and green asterisks indicate *CNGC* genes with high expression in pollen in *O. sativa* (in this study) and *A. thaliana*, respectively. (**B**) Schematic of the protein structure. The sequence of the protein domain scheme was arranged to be the same as the gene sequence of the phylogenetic tree of (**A**). The length of each CNGC protein is shown on the right; the unit is aa (amino acid). Blue boxes indicate the position and length of the conserved CT-CNBD (cyclic nucleotide-binding domain) of the CNGC family.

**Figure 2 plants-11-03145-f002:**
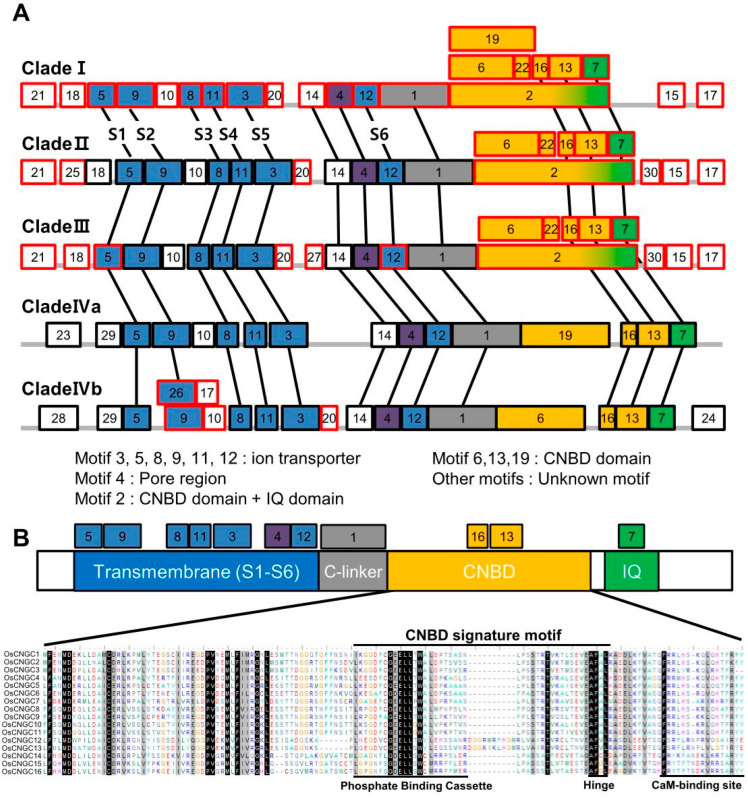
(**A**) MEME domain analysis and schematic diagram for main motif structures of five *CNGC* gene pairs from five CNGC subfamilies, respectively, in rice and Arabidopsis. The red and black squares borders represent non- and conservative MEME domains in the subfamily; their accumulation positions are the location of the corresponding amino acid positions. Motifs with known functions are marked in distinct colors (blue, purple, gray, yellow, orange, and green). Other colorless boxes indicate unknown motif. The known function domains were obtained from the SMART and PFAM tools. (**B**) Structural diagram of conserved MEME motifs in all CNGC proteins of rice and Arabidopsis. The transmembrane region, c-linker, CNBD domain, CaMBD domain, and IQ domain are represented by several rectangles with the colors of blue, gray, yellow, orange, and green, respectively. Below is the alignment of the CNBD domains of 16 CNGCs.

**Figure 3 plants-11-03145-f003:**
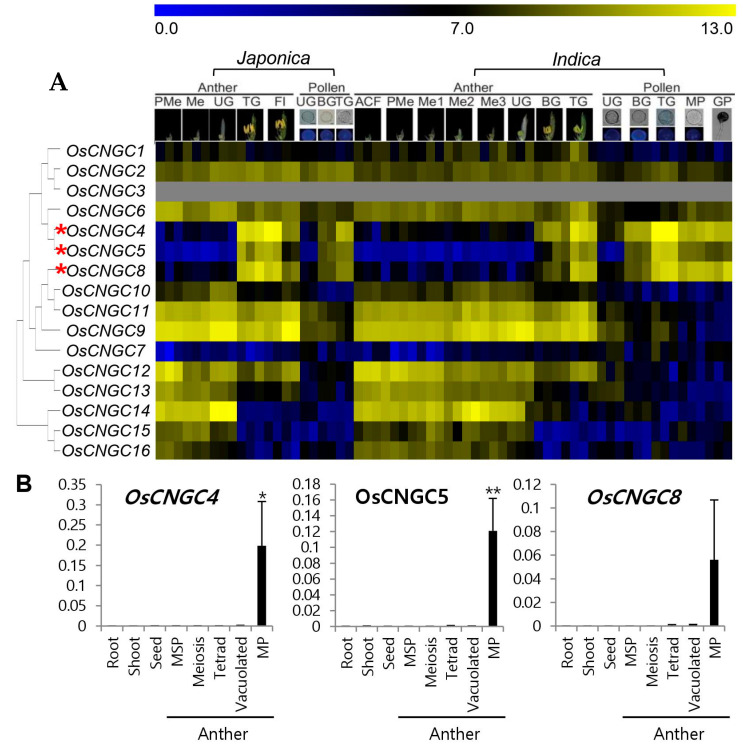
(**A**) Heatmap expression and phylogenetic analysis of 16 *OsCNGC* genes revealed three genes that were preferentially expressed in pollen. Yellow color in the heatmap indicates a high level of expression, whereas dark blue indicates low expression. Numeric values indicate an average of the normalized log2 intensity value of the microarray data. Pollen-preferentially expressed *OsCNGC* genes are indicated by red asterisks. ACF, archesporial cell-forming stage; BG, bicellular gametophyte stage; Fl, flowering stage; GP, germinated pollen; Me, meiotic stage; Me1, meiotic leptotene stage; Me2, meiotic zygotene-pachytene stage; Me3, meiotic diplotene-tetrad stage; MP, mature pollen stage; PMe, pre-meiosis; TG, tricellular pollen stage; UG, uni-cellular gametophyte stage. (**B**) Expression of pollen-preferred *OsCNGC* genes analyzed via qPCR in various tissues of rice. Msp, Microspore; MP, mature pollen. Rice ubiquitin 5 (*OsUbi5, LOC_Os01g22490*) was used as an internal control. The y-axis shows the expression level relative to *OsUbi5*, while the x-axis shows the samples used for analyses. Error bars represent the standard errors of three biological replicates. Significant differences are indicated by asterisks; * *p* < 0.01 and ** *p* < 0.0001.

**Figure 4 plants-11-03145-f004:**
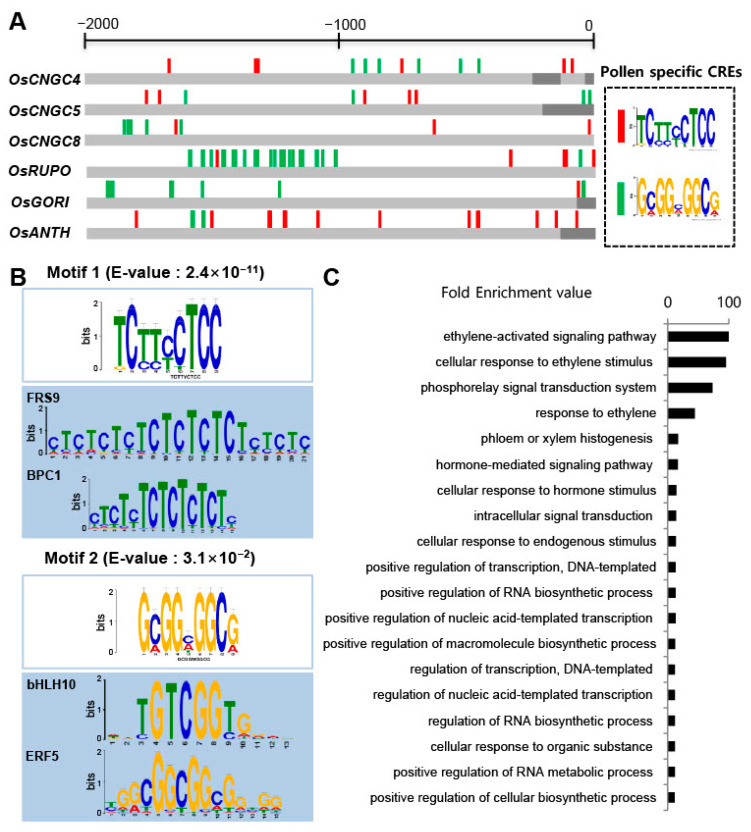
(**A**) Identification of putative cis-acting regulatory elements (CREs) found in pollen-preferential-expressed *OsCNGC* genes with *OsRUPO, OsGORI*, and *OsANTH* using the MEME suite. The number on the scale bars above the figure shows the upstream position of the promoter base pair when taken at +1 of ATG. Up to the upstream, 2000 base pairs were analyzed. Deep gray bars indicated the 5’-untranslated regions and brown bars indicate the promoter regions. (**B**) Sequence motifs for transcription factors with sequence similarity to pollen-preferential CREs. (**C**) A Gene Ontology (GO) enrichment analysis of 105 pollen-preferential CREs-associated genes. The 16 most significantly (*p* < 0.05) enriched GO terms in biological process, molecular function and cellular component branches are presented.

**Figure 5 plants-11-03145-f005:**
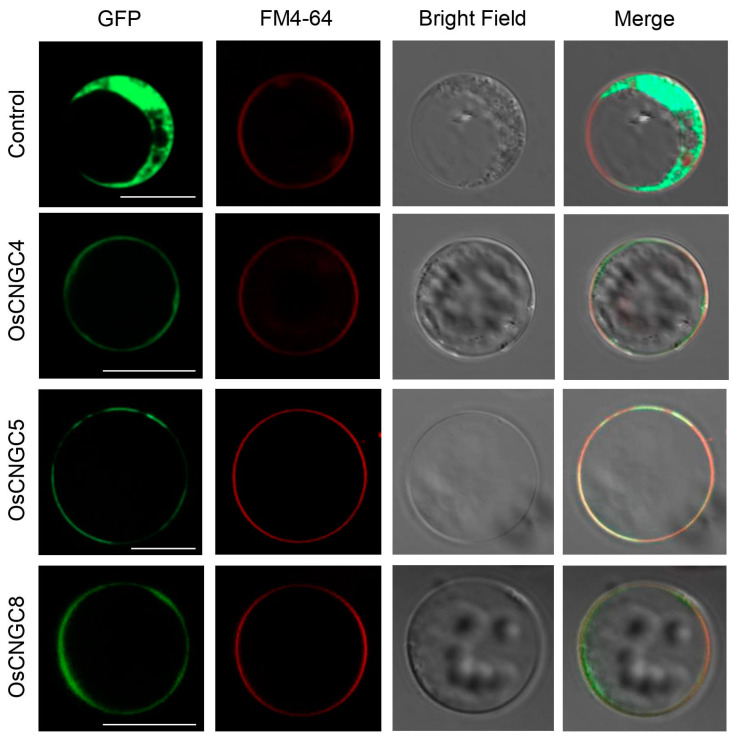
Three pollen-preferential expressed OsCNGC-green fluorescence protein (GFP) signals were observed in plasma membrane in rice protoplast. The second panel shows the membrane marker through FM4-64 staining, the third panel shows the bright-field image, and the last panel shows merged images of GFP and FM4-64. The red signal shows the membrane, the green signal shows OsCNGC localization, and the yellow signal shows merged images from the OsCNGC localization and PM signal. Scale bar = 20 μm. Experiments were repeated three times with similar results.

**Figure 6 plants-11-03145-f006:**
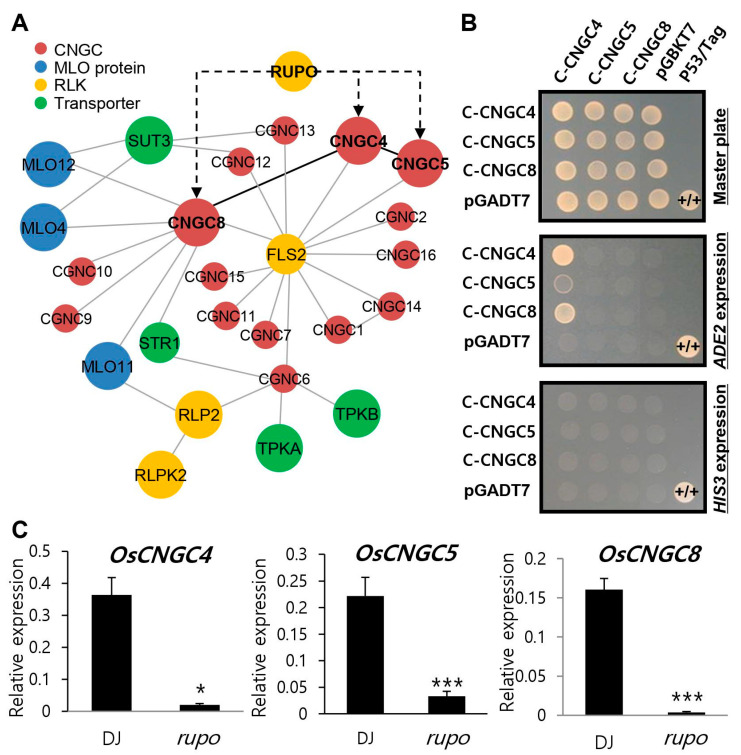
(**A**) The CNGCs network was drawn based on string database and yeast two-hybrid interaction data in this study. Red circles indicate OsCNGC proteins; blue circles indicate MLO proteins; yellow circles indicate RLKs (receptor-like-kinases); green circles indicate transporter proteins. Bold letters indicate OsCNGC with pollen-preferred expression. Black edges indicate physical interactions; gray edges indicate reference-based interaction; the dotted lines indicate indirect regulation. (**B**) C-CNGC4, C-CNGC5, and C-CNGC8. C-CNGC4, C-terminal region of OsCNGC4 from 552 aa to the end; C-CNGC5, C-terminal region of OsCNGC5 from 581 aa to the end; C-CNGC8, C-terminal region of OsCNGC8 from 430 aa to the end. AH109 yeast transformants were dropped onto selective medium lacking Leu and Trp (SD-LW) or also lacking Ade (SD-LWA) or Ade and His (SD-LWAH), with or without 5 mM 3-amino-1,2,3-triazole (3-AT), a competitive inhibitor of yeast HIS3. Positive control, yeast transformed with the P53 bait plasmid and the Tag prey plasmid. Negative control, yeast transformed with the parental bait vector (pGBKT7) and the prey vector (pGADT7). (**C**) Relative expression of pollen-preferred *OsCNGC* genes between DJ (wild-type rice) and knockout *rupo* mutant. Rice ubiquitin 5 (*OsUbi5, LOC_Os01g22490*) was used as an internal control. Error bars represent the standard errors of three biological replicates. Significant differences are indicated by asterisks, * *p* < 0.05. *** *p* < 0.001.

## Data Availability

All of the data provided in this study are available within this article.

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
