# Peer review of "Genome-Wide Analysis of Cyclic Nucleotide-Gated Channel Genes Related to Pollen Development in Rice"

_plants, 2022, doi:10.3390/plants11223145_

Round 1

Reviewer 1 Report

This is an interesting paper that combines bioinformatic and expression studies to make the case for three CNGC genes as components of rice pollen tube biology. Further experimental work, method details and reference updates are necessary, detailed as follows:

17 The claim here is for pollen-specific roles but in the rest of the MS it is clear that the genes in question are not specifically expressed in pollen. The text needs moderating.

24 The expression, not the activity can be modulated by RUPO.

45 Change nonselective to tetrameric. There isn’t enough evidence to claim that the whole family encodes for nonselective cation channels.

48 Ref 12 is out of date as a plant CNGC review. Perhaps acknowledge Fromm instead and cite a more recent review?

50 It is incorrect to say that “By contrast, in plants, the structures and functions of the CNGC family have only 50 recently been studied [7].” Additionally, the statement contradicts the rest of the paragraph that details structures. The choice here is to delete the statement or amend it to reflect the ongoing work, citing more recent studies on structure-function by the Luan, Yoshioka and Dietrich groups.

61 In particular, the cation channel in legumes have been used to elucidate the contribution of the CNGC family to the signal transduction pathway [26]. Channel and pathway should be plurals.

62 References for Arabidopsis CNGCs in pollen tube growth need to be updated to include Meng at minimum.

119 Method for anther wall extraction is needed.

322 “The OsCNGC genes are generally known to encode plasma membrane (PM) proteins that regulate the movement of nonselective cations.” There isn’t enough direct experimentation to support that statement and the term “nonselective cations” makes no sense. Please rephrase to “The OsCNGC genes are generally held to encode cation channel subunits. These may be expressed in the plasma membrane (PM).”

337 Fig 5 there is no GFP control shown. This requires correction. The GFP isn’t only seen at the PM but also in bodies beneath the PM for all 3 genes. This requires explanation. Number of trials needs to be given. Can the protoplasts be burst as a test of PM localisation? This seems essential as the claim is that the FM dye also stains the tonoplast.

355 Reference should be to 6A not 5A.

There is no methodology reported to support the section on signaling pathway construction. It must be provided.

362 yjr siganling? What does this mean?

357 It should be made clear what FLS2 is. RUPO should be introduced earlier and defined in this section prior to using the mutant otherwise the rationale for the experiment is not at all clear.

380 Refer to fig 6B not 5B. 385 Same problem for 5C.

386 The text reports that three genes were suppressed in the rupo mutant but only two are shown as significant in Fig. 6C.

392 Rephrase to “Plant CNGC proteins form tetrameric cation channels”

394 “…that were highly expressed in pollen, regulating late pollen development.” No evidence has been presented for these genes regulating pollen devlopement so this statement must be moderated.

451 Please cite the works of Yoshioka and Luan on heterotrimeric CNGC formation and the importance in signaling.

485 It needs to be made clear that pollen MLOs direct positioning of CNGCs and Meng et al. has to be cited (https://www.nature.com/articles/s41477-020-0599-1).

Throughout the MS, ionic valencies must be superscripted.

Author Response

L17 The claim here is for pollen-specific roles but in the rest of the MS it is clear that the genes in question are not specifically expressed in pollen. The text needs moderating.

Response: Following reviewer comment, we have corrected the sentence (Please see line 17).

L24 The expression, not the activity can be modulated by RUPO.

Response: We modified "activity" to "expression" (Please see line 24).

L45 Change nonselective to tetrameric. There isn’t enough evidence to claim that the whole family encodes for nonselective cation channels.

Response: As reviewer commented, we modified "nonselective" to "tetrameric" (Please see line 45).

L48 Ref 12 is out of date as a plant CNGC review. Perhaps acknowledge Fromm instead and cite a more recent review?

Response: We replaced it with a new reference (Please see line 48).

L50 It is incorrect to say that “By contrast, in plants, the structures and functions of the CNGC family have only 50 recently been studied [7].” Additionally, the statement contradicts the rest of the paragraph that details structures. The choice here is to delete the statement or amend it to reflect the ongoing work, citing more recent studies on structure-function by the Luan, Yoshioka and Dietrich groups.

Response: The sentence was revised by reflecting the recent reference you commented (Please see lines 50-51).

L61 In particular, the cation channel in legumes have been used to elucidate the contribution of the CNGC family to the signal transduction pathway [26]. Channel and pathway should be plurals.

Response: It was corrected in plurals (Please see lines 62-63).

L62 References for Arabidopsis CNGCs in pollen tube growth need to be updated to include Meng at minimum.

Response: We have updated to include Meng et al in the reference (Please see line 65).

L119 Method for anther wall extraction is needed.

Response: The sentence was corrected because anther wall sample was not used (Please see line 120).

L322 “The OsCNGC genes are generally known to encode plasma membrane (PM) proteins that regulate the movement of nonselective cations.” There isn’t enough direct experimentation to support that statement and the term “nonselective cations” makes no sense. Please rephrase to “The OsCNGC genes are generally held to encode cation channel subunits. These may be expressed in the plasma membrane (PM).”

Response: We have revised manuscript to be clearer by including reviewer’s commented sentence (Please see lines 330-331).

L337 Fig 5 there is no GFP control shown. This requires correction. The GFP isn’t only seen at the PM but also in bodies beneath the PM for all 3 genes. This requires explanation. Number of trials needs to be given. Can the protoplasts be burst as a test of PM localisation? This seems essential as the claim is that the FM dye also stains the tonoplast.

Response: We did the experiment again by adding GFP control, and adjusted excitation/emission wavelength for observation except chlorophyll (autofluorescence). As a result, the signals seen in the bodies beneath the PM of the GFP and FM4-64 channels were hardly visible. We modified Figure 5.

FM4-64 staining requires its internalization, so FM4-64 does not label the tonoplast alone. However, relative specific labeling is possible in a time-dependent manner: Long-term uptake (starting 2–3 h after application) will lead to a high accumulation of the dye at the tonoplast (Rigal et al., 2015).
We treated the protoplast with FM4-64 and observed the cells within 10 minutes. Therefore, it is more likely that the part dyed with FM4-64 is PM, not Tonoplast.

L355 Reference should be to 6A not 5A.

Response: we have corrected the error (Please see line 363).

There is no methodology reported to support the section on signaling pathway construction. It must be provided.

Response: A related method has been added (Please see lines 156-161).

L362 yjr siganling? What does this mean?

Response: We deleted the contents and explained the network (Please see line 371).

L357 It should be made clear what FLS2 is. RUPO should be introduced earlier and defined in this section prior to using the mutant otherwise the rationale for the experiment is not at all clear.

Response: Following the reviewer's comments, we clarified FLS2 and introduced RUPO (Please see lines 365-366, 392-400).

L380 Refer to fig 6B not 5B. 385 Same problem for 5C.

Response: we have corrected the error (Please see lines 389, 402).

L386 The text reports that three genes were suppressed in the rupo mutant but only two are shown as significant in Fig. 6C.

Response: Standard Deviation was reduced through 3 repeated re-experiments, and Figure was updated by adding the statistical significance (Please see Fig. 6C).

L392 Rephrase to “Plant CNGC proteins form tetrameric cation channels”

Response: We have revised manuscript as suggested (Please see line 407).

L394 “…that were highly expressed in pollen, regulating late pollen development.” No evidence has been presented for these genes regulating pollen devlopement so this statement must be moderated.

Response: Following reviewer comment, we have moderated the sentence (Please see lines 409-410).

L451 Please cite the works of Yoshioka and Luan on heterotrimeric CNGC formation and the importance in signaling.

Response: We cited the paper (Please see line 466).

L485 It needs to be made clear that pollen MLOs direct positioning of CNGCs and Meng et al. has to be cited (https://www.nature.com/articles/s41477-020-0599-1).

Response: We have revised the content by citing the paper (Please see line 497-498).

Throughout the MS, ionic valencies must be superscripted.

Response: we have corrected the error throughout the MS.

Reviewer 2 Report

The manuscript Plants-1930103 deals with very interesting and important topic of one group of genes controlling pollen development in rice. Excellent results and very good presentation makes the manuscript very suitable for publishing in Plants with a subject to minor revision and addressing the following points listed below, which I suppose cannot take too much time and attention of the authors.

Minor comments and corrections:

(1) L72-80, 87-95, 126, 193, 207, 378-383, and in many-many other parts of entire manuscript. This is not a mistake to write a text from the ‘first face’ using pronouns ‘we’. However, in the current manuscript, this is clearly over-excessive of the ‘first face’ writing and there are too many sentences with ‘we’. Authors must understand that this manuscript (and future published paper) is not a personal communication between authors and the Editor. Moreover, the general rule prescribes that manuscripts have to be written ‘from the third face’. For example, ‘Genome-wide analysis was performed …’; ‘Three pollen-preferred CNGC genes were identified …’; ‘A CNGC-mediated regulatory network was generated …’; and so on. Therefore, I am strongly asking authors to re-phrase almost all sentences with ‘we’, transforming it into writing from ‘third face’. The only exception can be made for very special cases like ‘We can hypothesize…’, where ‘first face’ writing is important. All other cases have to be modified.

(2) L101-102. The link to ‘Phytozome’ is incorrect and has a mistake in the end. It has to be ‘gov’ but not ‘g.,ov’.

(3) L119, L192, and maybe in other spots. Please check and correct all Botanical names of plant species, which must be written in Italics.

(4) L124-125. In the regime for qPCR, please add number of the used cycles.

(5) L291. The following phrase is very colloquial and has to be improved: “…CREs were sent to TomTom to find…”. Based on the information in L114, it might be like those ‘…sequences of CREs were analysed using TomTom tool to find…’.

(6) L372, 384, 385, and in other places. The names of mutants must be in Italics and starting from a small letter like those ‘…and knockout rupo mutant’.

(7) L383-384 and L393, and in many other places. Names of all genes must be in Italics as authors correctly used in L393. However, authors must be consistent and use the same font in Italics for all other genes (see L383-384). Authors must check and correct the entire manuscript for names of genes.

(8) L491-501 and in entire text of the manuscript. Authors indicated, provided a list and references in the text for Supplementary materials. However, no any Supplementary materials were found on the web-site of the submitted manuscript. In such conditions, I am unable to check and evaluate the Supplementary materials, and this point is remaining in the power and decision of the Handling Editor in this manuscript.

Author Response

(1) L72-80, 87-95, 126, 193, 207, 378-383, and in many-many other parts of entire manuscript. This is not a mistake to write a text from the ‘first face’ using pronouns ‘we’. However, in the current manuscript, this is clearly over-excessive of the ‘first face’ writing and there are too many sentences with ‘we’. Authors must understand that this manuscript (and future published paper) is not a personal communication between authors and the Editor. Moreover, the general rule prescribes that manuscripts have to be written ‘from the third face’. For example, ‘Genome-wide analysis was performed …’; ‘Three pollen-preferred CNGC genes were identified …’; ‘A CNGC-mediated regulatory network was generated …’; and so on. Therefore, I am strongly asking authors to re-phrase almost all sentences with ‘we’, transforming it into writing from ‘third face’. The only exception can be made for very special cases like ‘We can hypothesize…’, where ‘first face’ writing is important. All other cases have to be modified.

Response: The overall sentences in MS were modified according to the reviewer's comments.

(2) L101-102. The link to ‘Phytozome’ is incorrect and has a mistake in the end. It has to be ‘gov’ but not ‘g.,ov’.

Response: we have corrected error (Please see line 103).

(3) L119, L192, and maybe in other spots. Please check and correct all Botanical names of plant species, which must be written in Italics.

Response: we have corrected errors throughout the MS.

(4) L124-125. In the regime for qPCR, please add number of the used cycles.

Response: We added number of the cycle (Please see lines 125-127).

(5) L291. The following phrase is very colloquial and has to be improved: “…CREs were sent to TomTom to find…”. Based on the information in L114, it might be like those ‘…sequences of CREs were analysed using TomTom tool to find…’.

Response: Following reviewer comment, we have corrected the sentence (Please see lines 298).

(6) L372, 384, 385, and in other places. The names of mutants must be in Italics and starting from a small letter like those ‘…and knockout rupo mutant’.

Response: we have corrected errors throughout the MS.

(7) L383-384 and L393, and in many other places. Names of all genes must be in Italics as authors correctly used in L393. However, authors must be consistent and use the same font in Italics for all other genes (see L383-384). Authors must check and correct the entire manuscript for names of genes.

Response: we have corrected errors throughout the MS.

(8) L491-501 and in entire text of the manuscript. Authors indicated, provided a list and references in the text for Supplementary materials. However, no any Supplementary materials were found on the web-site of the submitted manuscript. In such conditions, I am unable to check and evaluate the Supplementary materials, and this point is remaining in the power and decision of the Handling Editor in this manuscript.

Response: Sorry to make it confuse. We submitted supplementary materials.